

# Identification of key miRNAs in the progression of hepatocellular carcinoma using an integrated bioinformatics approach

Qi Zheng[1], Xiaoyong Wei[2], Jun Rao[2] and Cuncai Zhou[2]

[1] Department of Oncology, Fuzhou First People's Hospital, Fuzhou, Jiangxi, China
[2] Department of Hepatobiliary Surgery, Jiangxi Cancer Hospital, Nanchang, Jiangxi, China

## ABSTRACT

**Background:** It has been shown that aberrant expression of microRNAs (miRNAs) and transcriptional factors (TFs) is tightly associated with the development of HCC. Therefore, in order to further understand the pathogenesis of HCC, it is necessary to systematically study the relationship between the expression of miRNAs, TF and genes. In this study, we aim to identify the potential transcriptomic markers of HCC through analyzing common microarray datasets, and further establish the differential co-expression network of miRNAs–TF–mRNA to screen for key miRNAs as candidate diagnostic markers for HCC.

**Method:** We first downloaded the mRNA and miRNA expression profiles of liver cancer from the GEO database. After pretreatment, we used a linear model to screen for differentially expressed genes (DEGs) and miRNAs. Further, we used weighed gene co-expression network analysis (WGCNA) to construct the differential gene co-expression network for these DEGs. Next, we identified mRNA modules significantly related to tumorigenesis in this network, and evaluated the relationship between mRNAs and TFs by TFBtools. Finally, the key miRNA was screened out in the mRNA–TF–miRNA ternary network constructed based on the target TF of differentially expressed miRNAs, and was further verified with external data set.

**Results:** A total of 465 DEGs and 215 differentially expressed miRNAs were identified through differential genes expression analysis, and WGCNA was used to establish a co-expression network of DEGs. One module that closely related to tumorigenesis was obtained, including 33 genes. Next, a ternary network was constructed by selecting 256 pairs of mRNA–TF pairs and 100 pairs of miRNA–TF pairs. Network mining revealed that there were significant interactions between 18 mRNAs and 25 miRNAs. Finally, we used another independent data set to verify that miRNA hsa-mir-106b and hsa-mir-195 are good classifiers of HCC and might play key roles in the progression of HCC.

**Conclusion:** Our data indicated that two miRNAs—hsa-mir-106b and hsa-mir-195— are identified as good classifiers of HCC.

Corresponding author
Xiaoyong Wei,
weixiaoyong206@163.com

## INTRODUCTION

Hepatocellular carcinoma (HCC) is one of the most common cancers worldwide and the second leading cause of global cancer-related death, accounting for around 11% of all cancer deaths (*Ferlay et al., 2015*). The pathogenesis of HCC, as well as the genes involved in this process is under intensive investigation. At present, the potentially curative options for HCC patients are radiofrequency ablation, liver transplantation and tumor resection. However, these options are only effective for the early stage of HCC (*Nault et al., 2018*). At the end stage of HCC, or when recurrence is occurred, the mean overall survival of patients are less than 3 months (*Yin et al., 2018*). It is necessary to understand the molecular mechanism of HCC progression, in order to predict and control this disease as early as possible.

MicroRNAs (miRNAs) are small non-coding RNAs that are highly conserved between species and are associated with many important biological processes. MiRNAs usually exert their roles in post-translational regulation, by base-pairing to the 3′-untranslated regions (3′-UTR) of their target mRNAs and suppressing the translation of target genes (*Bartel, 2004*). A single miRNA can target a broad range of mRNAs and suppress translation via non-perfect pairing with target mRNAs, causing degradation of target RNAs by the RISC. Recently, functional genomics studies have identified roles for miRNAs in the regulation of HCC development (*Boye & Yang, 2014*). MiRNAs are reported to regulate the proliferation, differentiation, apoptosis, invasion and metastasis of hepatoma cells through interacting and suppressing the functions of hepatic transcriptional factors (TFs) (*Lu et al., 2013*). Moreover, many HCC-related clinicopathological parameters have been linked to aberrant expression of miRNAs, including HBV/HCV status, survival, recurrence and metastasis formation. These features render miRNAs as suitable biomarkers for HCC diagnosis (*Klingenberg et al., 2017*). However, classical approaches of searching for miRNA biomarkers, including identification of differentially expressed genes (DEGs), may not reveal the complex interactions and functional association among genes in HCC development. Novel approaches of systematically studying the interaction network of protein-coding genes and miRNAs are still missing.

In this study, we aim to study the interactions of mRNA, miRNAs and TFs to reveal key miRNAs that are involved in HCC development, and to explore novel indicators of HCC progression. By establishing a weighed gene co-expression network and analyzing mRNA–miRNA–TF interactions, we identified two miRNAs—hsa-mir-106b and hsa-mir-195, that share the most TFs with their interacting mRNAs. Gene set enrichment analysis revealed that they are both involved in metabolic process and immune response. Our data indicated that hsa-mir-106b and hsa-mir-195 are good classifiers of HCC and key players in the progression of liver cancer.

## MATERIALS AND METHODS

The flowchart of identification of key mRNAs and mRNA-interacting miRNAs was shown in Fig. 1 and methods in detail are provided below.

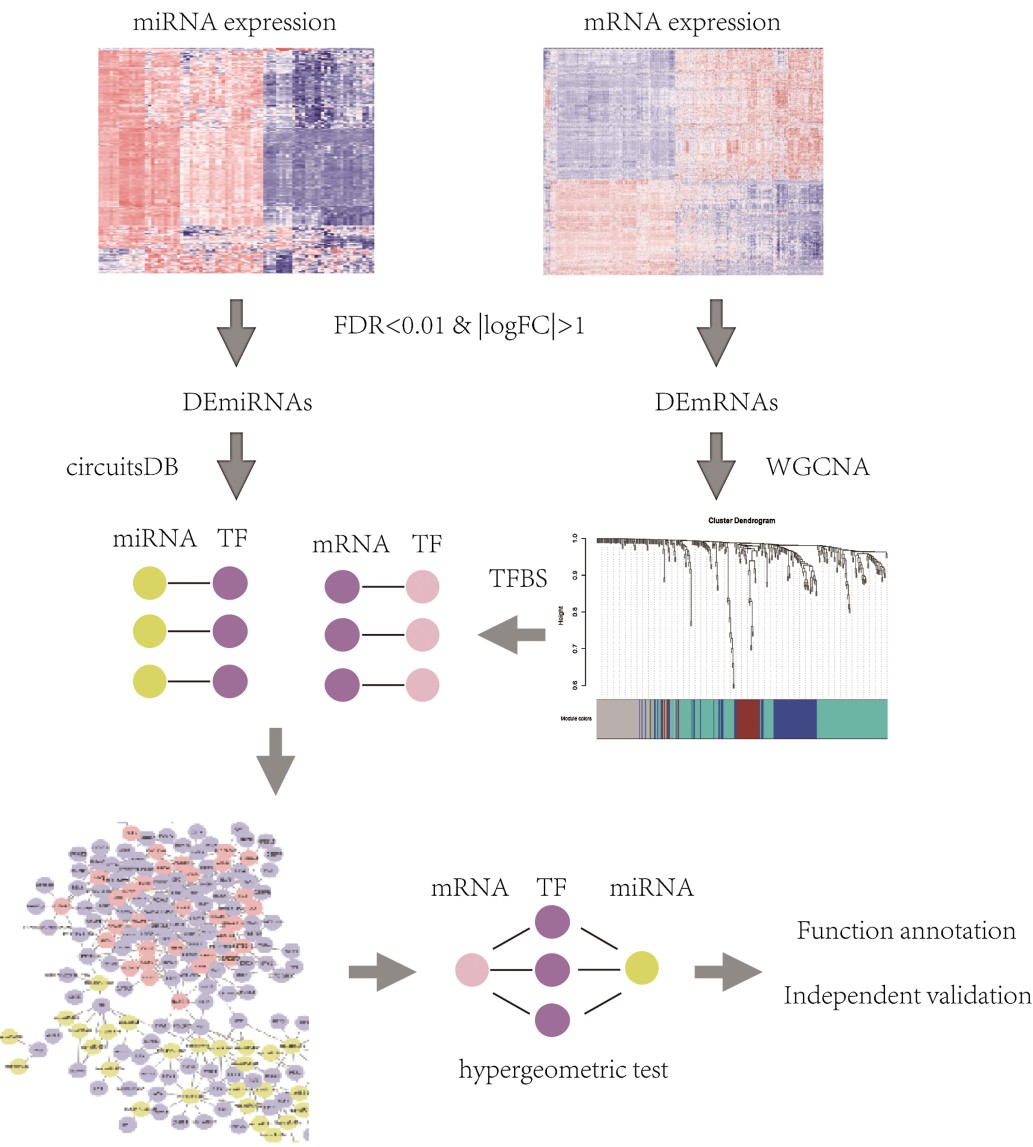

**Figure 1 The flowchart of identification of key mRNAs and mRNA-interacting miRNAs.**

## Dataset acquirement and processing

### Dataset acquirement

We searched for HCC related datasets in GEO database (https://www.ncbi.nlm.nih.gov/geo/) and downloaded two gene profiles in Series Matrix format. One dataset named GSE36376 is an encoded gene expression profile data containing 211 HCC samples and 243 adjacent non-tumor tissue samples, hybridized on Illumina HumanHT-12 V4.0 expression beadchip (platform no. GPL10558) (*Lim et al., 2013*). The other dataset named GSE36915 is a miRNA profile hybridized on Illumina Human v2 MicroRNA expression beadchip (platform no. GPL8179), containing 68 HCC samples and 21 non-tumor liver tissue samples (*Shih et al., 2012*).
*Dataset processing*

The dataset GSE36376 containing total of 47,323 probes and the dataset GSE36915 containing 1,145 probes were interpreted according to the following procedures: (1) In case that one probe corresponded to one gene uniquely, directly transformed the name of this probe into the corresponding gene symbol; (2) In case that one probe corresponded to multiple genes, removed this probe; (3) In case that multiple probes corresponded to one gene, transformed the mean value of these probes into the expression value of this gene. Altogether, we got the interpreted gene expression profile and miRNA profile containing 18,076 and 1,134 genes from the two datasets, respectively.

## Identification of DEGs

In order to identify genes (including mRNA and miRNA) that play key roles in HCC progression, R package Limma was used for *t*-test analysis and BH method was used for controlling the false discovery rate (FDR) and calculating the fold change of expression value. Genes with the FDR value less than 0.01 and the |logFC| value more than 1 was considered significantly DEGs.

## Construction of weighted gene co-expression networks and identification of gene modules associated with HCC progression

Weighed gene co-expression network analysis (WGCNA) is a freely accessible R package for the construction of weighted gene co-expression networks (*Langfelder & Horvath, 2008*). The basis of WGCNA is to find gene modules that have co-expression relationship when they are used to distinguish a sample trait in two groups. The procedure of WGCNA is composed of two main steps, expression clustering analysis and phenotypic association analysis, as described previously. Three different ways can be selected to construct the network and identify modules according to different needs. In our study, the one-step function was used for network construction and detection of differentially expressed mRNA modules.

## Prediction of mRNA–TF interaction

Interaction of mRNA and TF were predicted using TFBstools. Gene modules obtained from the weighted gene co-expression networks were combined with TFs in JASPAR database, and further analyzed for potential binding sites of TFs on mRNA molecules. The sequence information of mRNAs was obtained from UCSC database, and two thousand base-pairs (2k bps) upstream from the transcription start site (TSS) were considered TF binding sites. The scores of mRNA–TF pairs were ranked and used for further analyzed.

## Prediction of miRNA–TF interaction

To analysis the interaction of miRNA and TFs, predictive information of miRNA–TF pairs was downloaded from CricuitsDB database. The CricuitsDB database integrates genome-wide transcriptional and post-transcriptional regulatory networks based on bioinformatic gene sequence analysis, and it is designed to identify the complex

feedforward loop regulatory relationships of human miRNA–TF (*Friard et al., 2010*). To further enhance the reliability of the prediction, the miRNA–TF pairs were validated using two databases—TargetScan and TargetMiner (*Jeggari, Marks & Larsson, 2012*). The miRNA–TF pairs that were confirmed in at least one database and included in differentially expressed miRNAs (DEGs) were considered as differentially expressed miRNA–TF pairs.

## Construction of the mRNA–TF–miRNA triple network

The mRNA–TF–miRNA triple interaction network was constructed as described previously (*Wang et al., 2017*; *Zhao et al., 2018*). Briefly, differentially expressed miRNA–TF pairs and mRNA–TF pairs (scoring at top 5%) that screened out previously were integrated, and the interaction network was visualized using Cytoscape (*Shannon et al., 2003*). We analyzed the topological properties of the regulatory networks with Cytoscape plugin and identified hubs.

## Identification of miRNA–mRNA pairs using hypergeometric distribution

Interactions between miRNA and mRNAs in the triple network were evaluated based on Hypergeometric distribution (*Yang et al., 2015*). The formula of calculating significance is

$$p = 1 - \frac{\binom{k}{m}\binom{n-k}{N-m}}{\binom{n}{N}}$$

$N$ is the number of TFs, m is the number of mRNA-interacting TFs, $n$ is the number of miRNA-interacting TFs, $k$ is the number of TFs that both interact with mRNAs and miRNAs, and $p$ value is the significance of mRNA–miRNA interaction. $p < 0.05$ is considered significant.

## Functional enrichment analysis

Functional enrichment analysis of mRNA-interacting miRNAs were performed using R package clusterProfiler. GO-termed analysis of CC, MF and BP were used to interpret the function of miRNA–mRNA pairs in HCC progression. Hierarchical network diagram of GO-term analysis was generated using binGO of Cytoscape.

## Validation of key miRNAs in independent expression profiles

To test the classification capacity of selected miRNAs and mRNAs, an independent miRNA expression profile (GSE10694) was downloaded from GEO database and DEGs were screened out using R package Limma with the threshold of FDR < 0.05. DEGs screened out from GSE10694 were further compared with miRNAs that were filtered out in the triple network. The miRNAs located in the intersection of these two miRNA expression profiles were considered key miRNAs that might drive HCC development.

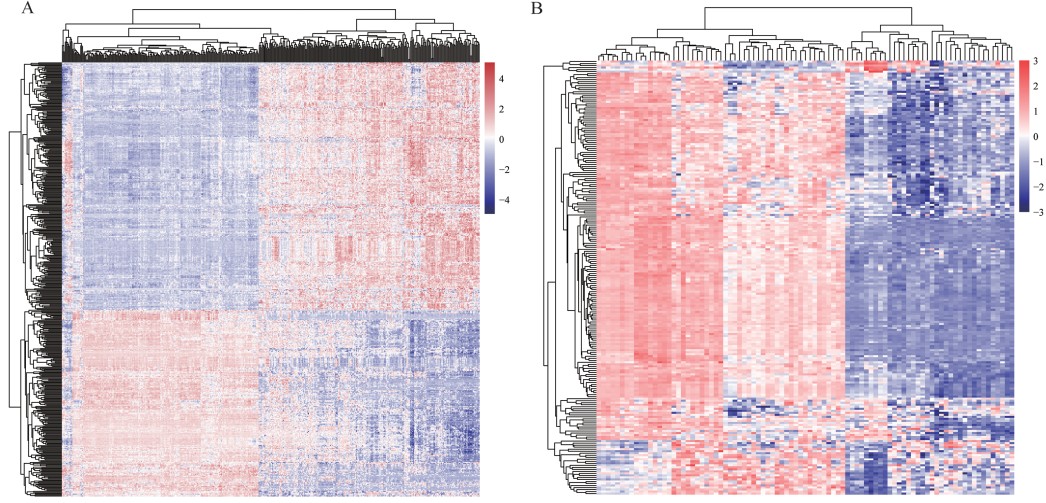

**Figure 2 Heatmaps of profiles GSE36376 (A) and GSE36915 (B) from GEO database.** The degree of expression is indicated by different colors, with expression increasing between blue and red. Blue, low expression; red, high expression.

# RESULTS

## Identification of DEGs from the expression profile of HCC

In order to find differentially expressed mRNAs and miRNAs between HCC and non-tumor controls, two expression profiles—one mRNA profile GSE36376 and one miRNA profile GSE36915, were downloaded from GEO database, respectively. Totally, GSE36376 contained 18,076 mRNAs, while GSE36915 contained 1,134 miRNAs. DEGs, including 465 differentially expressed mRNAs between 211 HCC tissues and 243 non-tumor tissues and 215 differentially expressed miRNAs between 68 HCC tissues and 21 non-tumor tissues, were identified using *t*-test analysis and FDR correction (listed in Table S1). Heatmaps of two profiles, as shown in Figs. 2A and 2B, respectively, indicated that these DEGs were good in distinguishing HCC samples and non-tumor samples.

## Construction of weighed gene co-expression network

Weighted gene co-expression networks are used to identify interesting gene modules, as well as the intramodular connectivity and gene significance based on the correlation of a gene expression profile with a sample trait. To explore the interaction of differentially expressed mRNAs, weighted gene co-expression network containing 465 mRNAs was constructed by WGCNA. The distributions of log(k) and log(p(k)) coefficients and mean connectivity corresponding to thresholds of significance (β) were computed by pickSoftThreshold and shown in Fig. 3A. Based on a scale-free topology criterion, β = 12 was selected as the soft threshold power in the present study. To obtain the co-expression matrix, the minimal module size was set as 20, and other parameters were set at default levels. The system clustering tree was constructed base on Pearson's coefficients of each gene (Figs. 3B and 3C). Therefore, genes were classified into three modules according to their correlation with traits. Among these, we found that the brown module eigengene, which includes 33 mRNAs (listed in Table S2), displayed highest correlation index with

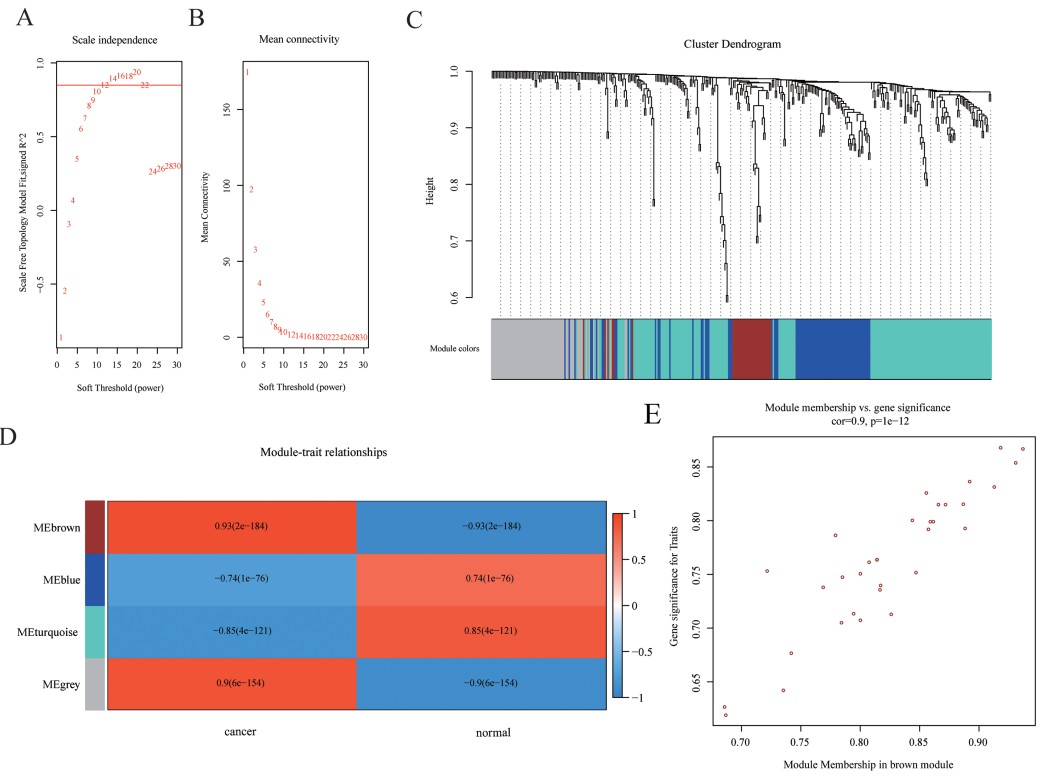

**Figure 3 Construction of WGCNA network.** (A–B) The distributions of log(k) and log(p(k)) coefficients (A) and mean connectivity (B) corresponding to thresholds of significance (β) were shown. (C) Construction of the system clustering tree based on Pearson's coefficients of DEGs. (D) Correlation between significance and gene expression of brown module membership. (E) Correlation between gene expression in modules and traits.

traits ($R$ = 0.93, $p$ = 2E−184, Fig. 3D). Intramodular interaction of these 33 mRNAs were analyzed and shown in Fig. S1. Expression of these mRNAs was highly correlated with traits (Fig. 3E). Taken together, we found a co-expression gene hub containing 33 mRNAs that were differentially expressed in HCC samples compared with controls.

## Prediction of the mRNA–TF interactions

To further explore potential interactions between mRNAs and TFs, we scored the binding of 145 TFs (downloaded from JASPAR database, and listed in Table S3) to the 33 mRNAs identified above using TFBstools. The sequence information of mRNAs was downloaded from USCS database, and sequences in the range of 2kb upstream from the TSS were picked up for predicting TF-binding sites. Top 5% mRNA–TF pairs were selected for further analysis. Totally, 32 mRNAs, 85 mRNA-interacting TFs, and 256 mRNA–TF pairs were identified (listed in Table S4).

## Prediction of interactions of miRNAs and TFs

The generation of miRNAs were regulated by TFs, while miRNAs were usually involved in post-translational regulation of TFs. Therefore, we predicted miRNA-interacting TFs by combining the information of three databases—CircuitsDB, TargetScan and TargetMiner.

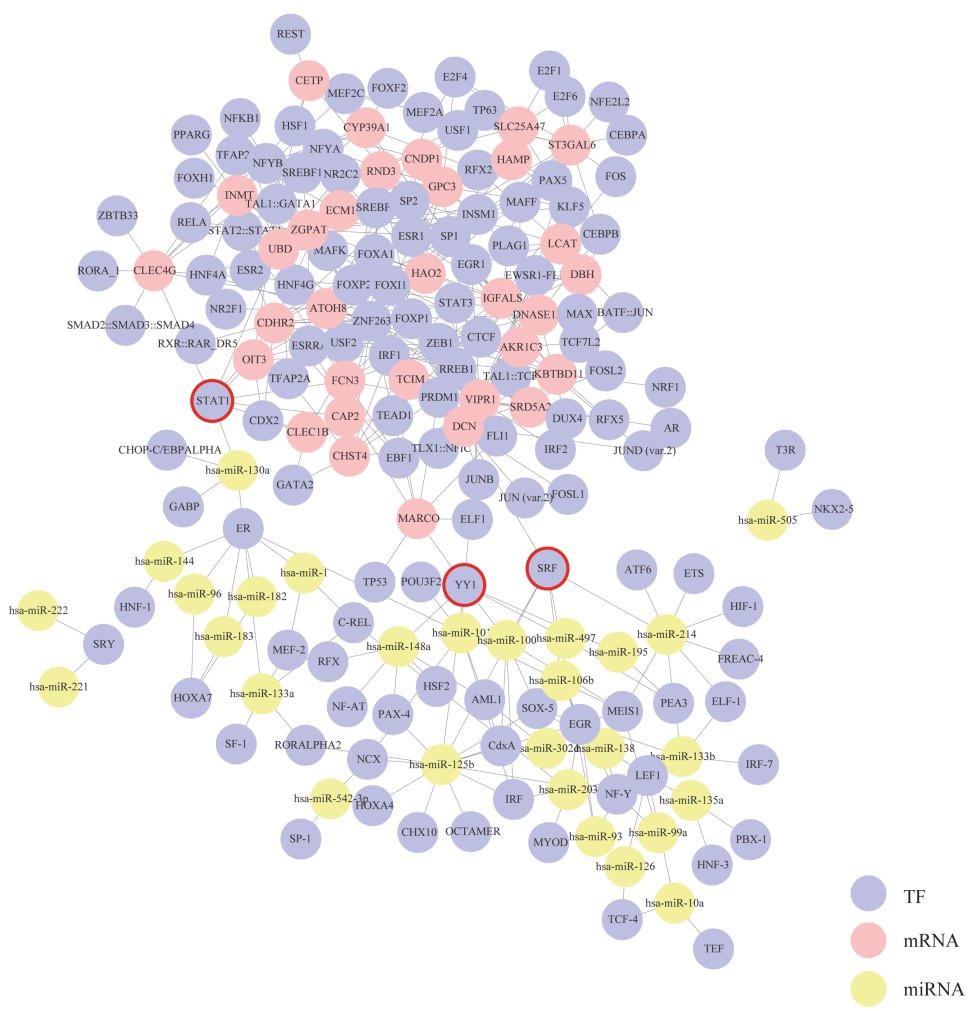

**Figure 4 Construction of the mRNA–TF–miRNA triple interaction network.** Each circular represents a gene. The purple circular represents TF, the red circular represents mRNA, and the yellow circular represents miRNA. Their interactions are displayed as solid lines. Core TFs that simultaneously connect with both mRNAs and miRNAs are emphasize with red circles.

TFs that interact with differentially expressed miRNAs were obtained from CircuitsDB, and miRNA–TF pairs were subsequently validated in TargetScan and TargetMiner. Totally, 100 miRNA–TF pairs containing 28 miRNAs and 45 TFs were identified (Table S5).

## Generation of mRNA–TF–miRNA interaction network

Based on the mRNA–TF pairs and miRNA–TF pairs identified above, a triple interaction network including mRNAs, TFs and miRNAs was generated (Fig. 4). This network contained 356 edges, 32 mRNA hubs, 28 miRNA hubs and 127 TF hubs. We found that three TFs—STAT1, SRF and YY1 were special because they were simultaneously connected with both mRNAs and miRNAs. According to our results, these three TFs might play critical roles in the progression of HCC.

**Table 1 List of miRNA–mRNA interactions.**

| miRNA | mRNA | | | | | | | | | | |
|---|---|---|---|---|---|---|---|---|---|---|---|
| hsa-miR-148a | MARCO | TCIM | | | | | | | | | |
| hsa-miR-302d | CAP2 | ECM1 | HAMP | TCIM | | | | | | | |
| has-miR-1 | CAP2 | TCIM | | | | | | | | | |
| hsa-miR-133a | TCIM | | | | | | | | | | |
| hsa-miR-130a | CAP2 | FCN3 | UBD | OIT3 | CLEC1B | TCIM | CLEC4G | | | | |
| hsa-miR-93 | DCN | CAP2 | TCIM | | | | | | | | |
| hsa-miR-106b | DCN | CAP2 | TCIM | | | | | | | | |
| hsa-miR-133b | TCIM | | | | | | | | | | |
| hsa-miR-96 | CAP2 | ECM1 | HAMP | TCIM | | | | | | | |
| hsa-miR-144 | CAP2 | ECM1 | HAMP | TCIM | | | | | | | |
| hsa-miR-183 | CAP2 | ECM1 | HAMP | TCIM | | | | | | | |
| hsa-miR-182 | CAP2 | ECM1 | HAMP | TCIM | | | | | | | |
| hsa-miR-135a | CAP2 | TCIM | | | | | | | | | |
| hsa-miR-138 | CAP2 | TCIM | | | | | | | | | |
| hsa-miR-542-3p | CAP2 | ECM1 | HAMP | TCIM | | | | | | | |
| hsa-miR-497 | VIPR1 | CAP2 | ECM1 | MARCO | HAMP | TCIM | | | | | |
| hsa-miR-195 | VIPR1 | CAP2 | ECM1 | MARCO | HAMP | TCIM | | | | | |
| hsa-miR-222 | DBH | CAP2 | RND3 | ECM1 | HAO2 | CNDP1 | CETP | OIT3 | CLEC1B | SRD5A2 | HAMP | TCIM |
| has-miR-221 | DBH | CAP2 | RND3 | ECM1 | HAO2 | CNDP1 | CETP | OIT3 | CLEC1B | SRD5A2 | HAMP | TCIM |
| hsa-miR-10a | CAP2 | TCIM | | | | | | | | | |
| hsa-miR-203 | TCIM | | | | | | | | | | |
| hsa-miR-100 | MARCO | TCIM | | | | | | | | | |
| hsa-miR-126 | CAP2 | ECM1 | HAMP | TCIM | | | | | | | |
| hsa-miR-99a | CAP2 | ECM1 | HAMP | TCIM | | | | | | | |
| hsa-miR-505 | CAP2 | ECM1 | HAMP | TCIM | | | | | | | |

### Refining mRNA–miRNA pairs in the triple network

To further identify core mRNA–miRNA interactions in the triple network, we used hypergeometric distribution method to evaluate the connectivity of mRNA–miRNA pairs. The degree of connectivity of mRNA–miRNA pairs were measured by the number of TFs they shared. Totally 100 mRNA–miRNA pairs, including 18 mRNAs and 25 miRNAs, were found to share one or more TFs (Table 1). According to our results, one miRNA interacts with several mRNAs while one mRNA interacts with several miRNAs.

### Functional enrichment analysis of mRNA-interacting miRNAs

The complex interactions of mRNAs and miRNAs indicated the biological roles of miRNAs in the progression of HCC. To further study the biological functions of 25 miRNAs identified above, we performed GO-term enrichment analysis, which included cellular component, molecular function (MF) and biological process (BP). The result of functional enrichment analysis indicated that these miRNAs were involved in the BPs such as negative regulation of vascular endothelial growth factor signaling pathway,

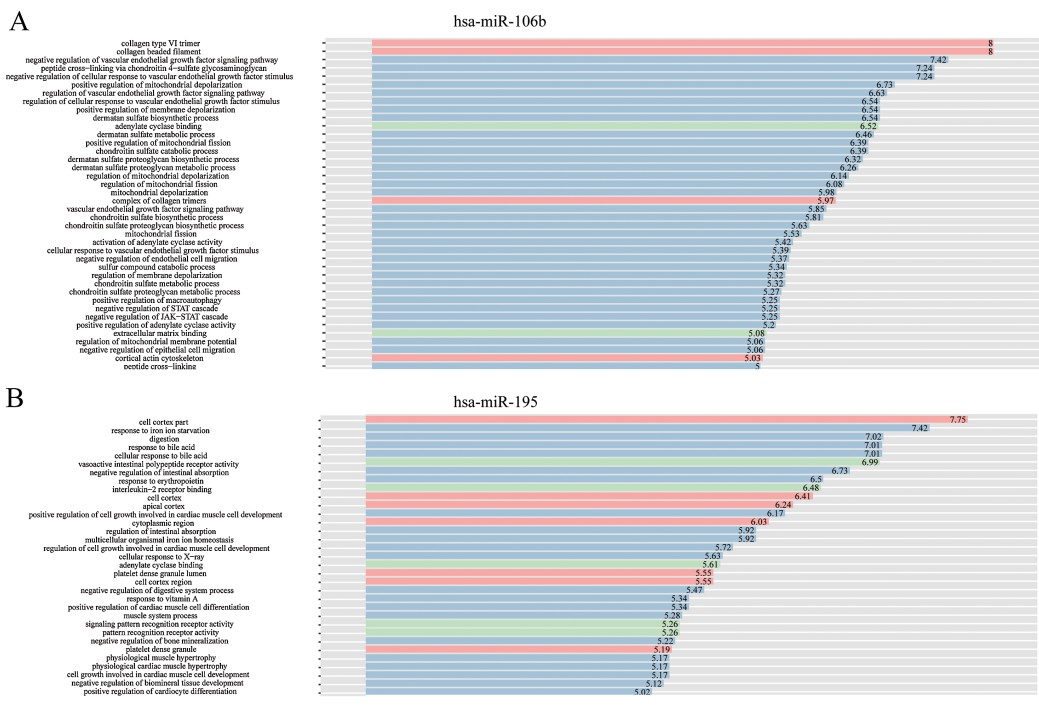

**Figure 5** **GO-term analysis of hsa-miR-106b and hsa-miR-195.** Cellular component (CC) of hsa-miR-106b (A) and hsa-miR-195 (B) are shown in red, biological process (BP) are shown in blue, and molecular function (MF) are shown in green. Description of GO-terms are listed.

response to iron on starvation, response to bile acid and urate homeostasis, and MFs such as adenylate cyclase binding, signaling pattern recognition receptor activity, and CCs such as collagen type VI trimer, cell cortex part (Fig. S2). Representative results of the GO-term analysis were shown in Fig. 5. GO-term analysis was further visualized by the hierarchical network diagram, which interpreted the weight and the hierarchical relationship of enriched GO-terms. The analysis results of two of the miRNAs— hsa-miR-106b and hsa-miR-195 indicated that they are intensely involved in aminoglycan metabolic process, peptide cross-linking, positive regulation of cyclase activity and immune system process, immune response (Fig. 6). Taken together, results of functional enrichment analysis indicated that most of the mRNA-interacting miRNAs that identified in the network might play important role in the progression of HCC.

## Verification of expression of miRNAs in another independent case

To further verify the reliability of the results, we downloaded an independent expression profile GSE10694 from GEO database. GSE10694 contains miRNA expression profiles in liver cancer tissues and corresponding non-tumor tissues of 78 HCC cases, and in 10 normal liver tissues (*Li et al., 2008*). Totally, 23 differentially expressed miRNAs were identified using Limma with the threshold of FDR < 0.05. Among these miRNAs, miR-106b and hsa-miR-195 were selected twice in GSE10694 and GSE36915, successively. This result indicated that the two miRNAs are good classifiers of HCC and key players in
A

B

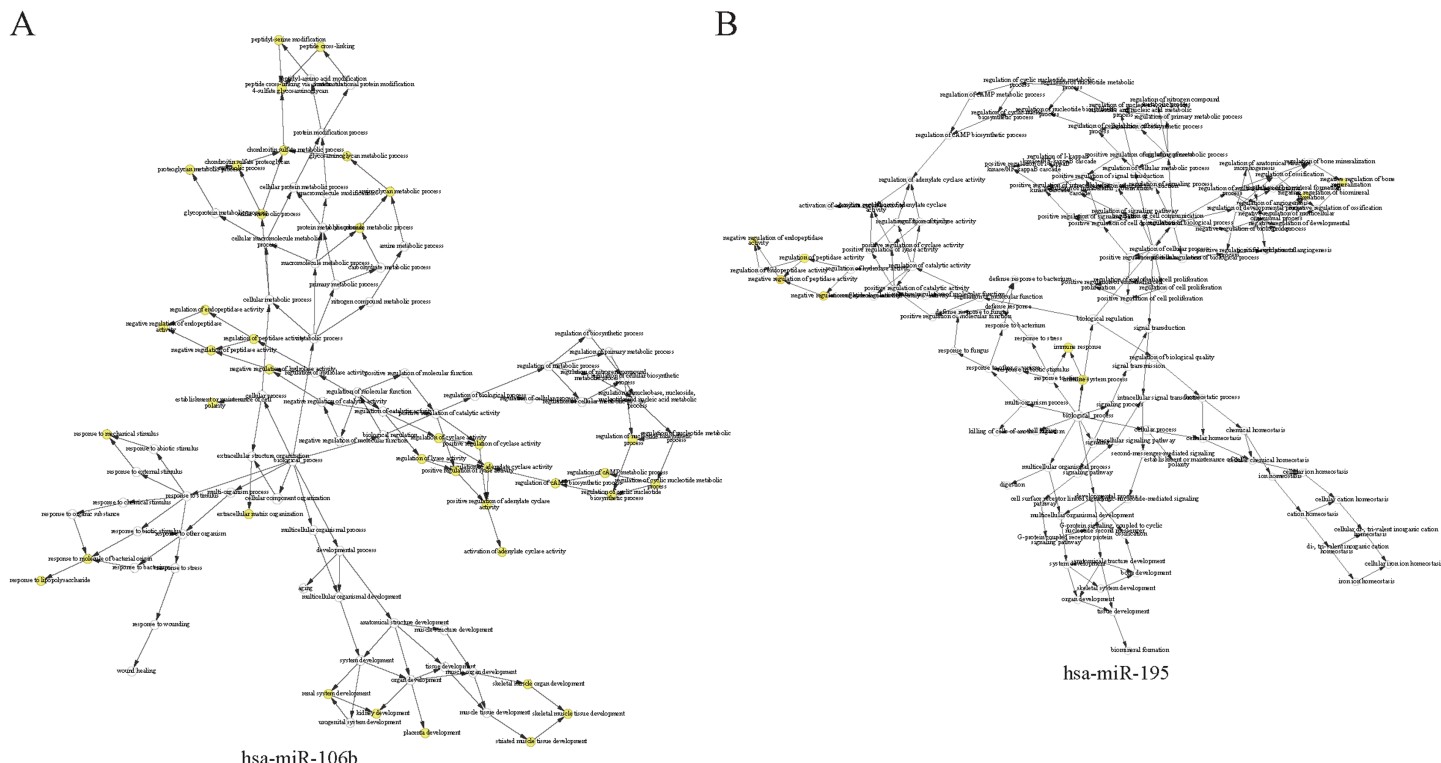

hsa-miR-106b

hsa-miR-195

**Figure 6 The hierarchical network diagram of GO-term analysis of hsa-miR-106b (A) and hsa-miR-195 (B).** The hollow circulars are GO terms. GO terms with high significance are colored with yellow.

the progression of liver cancer. On the other hand, our results also confirmed that the method used to identify these miRNAs are reliable and can be applied in other cases.

## DISCUSSION

1. In this study, we used deep and comprehensive bioinformatic analysis of three independent public datasets and identified a considerable number of mRNA–miRNA pairs. Moreover, our study utilized the central regulatory roles of TFs to construct a triple interaction network that can systematically interpret the relationship between miRNAs, TFs and mRNAs. TFs are key modulators of hepatic functions and are broadly involved in HCC development because of their DNA binding and transcriptional modulating activity. We found that 25 miRNAs and 18 mRNAs among these mRNA–miRNA pairs shared the most TFs and located in the central position of the miRNA–TF–mRNA triple interacting network. Therefore, we succeeded to predict that these mRNA–miRNA pairs might play central roles in the development of HCC and provide a novel way to explore differentially co-expression gene hubs and interpret their interactions. These co-expression gene hubs are valuable and can be used as diagnostic biomarkers for HCC, although more experimental verification of the results needed to be performed in the future.

2. Among the genes identified in the triple network, some have been previously connected to cancer progression in independent studies. For instance, signal transducer and

activator of transcription 1 (STAT1), a transcriptional factor associated with type I and II interferon signaling, is considered to have anti-tumor activity in the HCC models. STAT1-deficient mice are more prone to tumor development than animals with wild-type, indicating the central role of STAT1 in HCC progression (*Shankaran et al., 2001*). Serum response factor (SRF) is a transcription factor that in liver, can be activated by hepatitis B virus (HBV) and C virus (HCV). Constitutive activation of SRF in mice contributes to primary HCC formation (*Ohrnberger et al., 2015*). In addition to TFs, other genes that are involved in immune response (MARCO) (*Sun et al., 2017*), iron homeostasis regulation (HAMP) (*Ren et al., 2018*), proteoglycans synthesis (ECM1 (*Chen et al., 2011*), DCN (*Horvath et al., 2014*)) have been reported to be associated with the progression or prognosis of HCC. Especially, two miRNAs—hsa-miR-106b and hsa-miR-195, which were verified as HCC classifiers in two independent profiles, have been also implicated in regulating HCC development. Hsa-miR-106b regulates the apoptosis and tumorigenesis of HCC via targeting Zbtb7a (*Liang et al., 2018*), while hsa-miR-195 inhibits metastasis of HCC via targeting FGF2 and VEGFA (*Wang et al., 2015*). Moreover, hsa-miR-195 has also been implicated to target PCMT1, a member of the type II class of protein carboxyl methyltransferase enzymes, thus regulating protein metabolism and facilitating the development of HCC (*Amer et al., 2014*). Therefore, all these reports solidly confirm that the genes selected in this hub are closely correlated with the progression of HCC. However, because only three expression profiles have been used in this study, the reliability of results need to be confirmed in more independent cases.

## CONCLUSIONS

In this study, we used an integrated bioinformatics approach to explore mRNA–miRNA–TF triple interactions that are highly associated with HCC development. Our data indicated that two miRNAs—hsa-mir-106b and hsa-mir-195 are identified as good classifiers of HCC. However, because miRNAs can directly bind to mRNAs and suppress their translational function, further studies should be performed to analysis the suppressive effect of miRNAs to mRNAs and TFs in this network. Moreover, before these results being applied in exploring clinical biomarkers or therapeutic targets, experimental studies should be performed to demonstrate the functional roles of these genes in different stages of HCC progression, respectively.

### Funding

This study was sponsored by the Key Research and Development Program of Jiangxi Province (20161ACG70016). The funders had no role in study design, data collection and analysis, decision to publish, or preparation of the manuscript.

## Grant Disclosures

The following grant information was disclosed by the authors:

Key Research and Development Program of Jiangxi Province: 20161ACG70016.

## Competing Interests

The authors declare that they have no competing interests.

## Author Contributions

- Qi Zheng conceived and designed the experiments, performed the experiments, authored or reviewed drafts of the paper, and approved the final draft.
- Xiaoyong Wei analyzed the data, prepared figures and/or tables, authored or reviewed drafts of the paper, and approved the final draft.
- Jun Rao performed the experiments, analyzed the data, prepared figures and/or tables, and approved the final draft.
- Cuncai Zhou conceived and designed the experiments, prepared figures and/or tables, and approved the final draft.

## Data Availability

The raw data is available at GEO: GSE36376, GSE36915 and GSE10694.

## Supplemental Information

Supplemental information for this article can be found online at http://dx.doi.org/10.7717/peerj.9000#supplemental-information.

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
