# Peer review of "Identification of key miRNAs in the progression of hepatocellular carcinoma using an integrated bioinformatics approach"

_PeerJ, doi:10.7717/peerj.9000_

## Round 0.1 · original submission · Major Revisions

· Academic Editor

Major Revisions

In this manuscript, Zheng et al. described the application of bioinformatics approaches to identify critical miRNAs involved in the progression of hepatocellular carcinoma. Two specialists in the field evaluated this submission, and both reviewers have concerns regarding the analysis of the results and the description of the methodology. Considering the evaluation of the reviewers, I recommend major revision in the manuscript.

Reviewer 1 ·

Basic reporting

1,"At present, the potentially curative options for HCC patients are radiofrequency ablation, liver transplantation and tumor resection. However, these options are only effective for the early stage of HCC." please add your referred literature. as i know radiofrequency ablation and tumor resection were also recommended for advanced stage HCC of specific patients in the guideline of HCC in 2017.
2, Please display the complete process of construction of the mRNA-TF-miRNA triple network. It's too general
3, where is the formula of calculating significance from? in which article, please add the literature; if created by yourself, please make some explanation to convince me.

Experimental design

how do you evaluate the reliability of the tool for prediction of miRNA-TF interaction?

Validity of the findings

I think that the miRNA-TF pairs that were confirmed in at least one database could not convinced me, I think it must be confirmed in both database.

Additional comments

please describe the Materials & Methods in details, especially mRNA-TF-miRNA triple network and Validation of mRNA-miRNA pairs.

·

Basic reporting

The authors used clear and unambiguous English, literature references are cited properly.
The Discussion part of the article is questionable, because the first paragraph represents combined text from Introduction and Results.
Some figures should be improved. For example, fig.6 has unreadable format

Experimental design

The main weakness of the study is the absence of the experimental verification of results obtained. Nevertheless, such deep and comprehensive bioinformatic analysis of public datasets is valuable and let to reveal prognostic biomarkers for HCC.
The strength of the study is in analysis on two large datasets with subsequent verification the results in independent dataset. Another advantage is in included TFs. The ternary mRNA-miR-TF networks are of certain interest in the HCC diagnostic.

Validity of the findings

The results obtained are robust and statistically sound. However the interpretation might be more meaningful. For example, it remains unclear if has-miR-106b and has-miR-195 bind to conservative sites. You can also specify if they abberantly expressed in other types of cancer.

Additional comments

My major suggestion concerns:
fig.1- please, suggest this figure in the schematic view for the better comprehension
fig.6 - please, revise the scale of the figure, at the moment it is uninformative
lines 236-242 the text repeats the Introduction part. Please, revise this paragraph
lines 242-246 there is one more repeating of the Results, please write more interpretation
line 27: rewrite mRNA-TF-microRNA

---

## Round 0.2 · accepted · Accept

· Academic Editor

Accept

The authors carried out all modifications indicated by the reviewers. The revised version of the manuscript improved a great deal and can be accepted as it is.

·

Basic reporting

The authors addressed all comments, which I gave in the first round of review. So, the manuscript meets the basic requirements

Experimental design

The authors made the necessary changes

Validity of the findings

The bioinformatic findings are valuable and let to reveal prognostic biomarkers for HCC. The subsequent experimental studies will verify the results obtained it this analysis.

Additional comments

The authors made the necessary changes in the revised version of the manuscript.